# A Hybrid Data-Differencing and Compression Algorithm for the Automotive Industry

**DOI:** 10.3390/e24050574

**Published:** 2022-04-19

**Authors:** Sabin Belu, Daniela Coltuc

**Affiliations:** Doctoral School of Electronics, Telecommunications and Information Technology, Politehnica University of Bucharest, 061071 Bucharest, Romania; sabin24@hotmail.com

**Keywords:** big data, automotive, delta encoding, bsdiff, vcdiff, OTA software update, Lempel–Ziv, LZ77, LZFG, LZMA

## Abstract

We propose an innovative delta-differencing algorithm that combines software-updating methods with LZ77 data compression. This software-updating method relates to server-side software that creates binary delta files and to client-side software that performs software-update installations. The proposed algorithm creates binary-differencing streams already compressed from an initial phase. We present a software-updating method suitable for OTA software updates and the method’s basic strategies to achieve a better performance in terms of speed, compression ratio or a combination of both. A comparison with publicly available solutions is provided. Our test results show our method, Keops, can outperform an LZMA (Lempel–Ziv–Markov chain-algorithm) based binary differencing solution in terms of compression ratio in two cases by more than 3% while being two to five times faster in decompression. We also prove experimentally that the difference between Keops and other competing delta-creator software increases when larger history buffers are used. In one case, we achieve a three times better performance for a delta rate compared to other competing delta rates.

## 1. Introduction

The exponential increase in data, also known as big data, within the last decade has made once-popular data compression unable to fulfill its basic tasks. Compressing big data to achieve a workable or more feasible form for easier storage or transfer is now one of the challenges of the century. New technologies are needed to address it, and delta encoding seems to be one of them. Delta encoding is a concept that tries to fill in gaps by recording only data that has changed between two versions of the same ‘object,’ e.g., a file, a buffer, or a stream of bytes. This recorded information on differences can be further encoded with an entropy coder or further compressed by using a fully-fledged dictionary-based compressor.

In many domains, and, in particular, in the automotive industry, huge amounts of time and resources are invested in testing to fulfill all specifications and requirements and comply with all regulations. Any major or minor software-release version is coupled with huge test logs and test result catalogs containing Excel sheets, files, screenshots, test cases, action step recordings, etc. Thus, software projects are becoming more and more complex every day, and developers must handle hundreds of thousands of lines of code per module or component on average. Nonetheless, software bugs and glitches tend to occur despite all the effort put into testing phases. Moreover, perhaps more than any other domains—except aviation—automotive software is supposed to be 100% reliable in any situation, from performing an e-call from the roadside following the deployment of an airbag to emergency braking to prevent an imminent collision or alert drivers of a vehicle’s presence in a blind spot. The best software reliability can be achieved only by staying up-to-date, and every time a software glitch has been detected and fixed, the ‘old’ software must be updated or patched. Usually, software fixes are prioritized and deployed through a unified system to each registered device linked to a specific vehicle series or type that is supposed to receive a certain update or patch.

Most vehicle OEMs (Original Equipment Manufacturers) issue software updates periodically for a variety of reasons other than bug fixes. For instance, quality patches are regularly added to improve overall performance, ranging from updating the infotainment device(s) for a better user experience inside the vehicle, to security-related updates, such as air-bag deployments or gas -consumption check-up software.

These software updates are must-haves if vehicle computers—and, by default, auto-mobiles—are meant to function at their expected qualities.

The automotive industry creates and uses various types of data, which may include portions of compressed streams. Simply using classical data compression is not sufficient when dealing with such data, which are commonly embedded. For example, consider PDX files, which are mostly collections of ZIP [1] files intermixed with other data and are re-zipped together later. The process can be repeated or combined many times. These files are extremely hard to compress since they may have already been packed in different ‘onion’-shaped layers of compressed streams. Delta encoding is the only solution to this problem since even if the source and target streams are compressed, there is a chance these streams may share identical portions, depending on the intermix of similar data between the source and the target files or the data streams included.

Few publicly available solutions for delta encoding and binary differencing are worth mentioning. The secure delta binary-differencing engines [2]—developed by agersoftware in collaboration with NetLUP Xtreme Technologies— and XtremeDELTA [3]—solely developed by NetLUP—are two of them. The two companies have developed an in-house solution that implements a hybrid technology extremely suitable for embedded platforms as well as any operating system due to a light decoding footprint; these engines use only a few megabytes of memory for decoding, and they are highly effective with very large files due to fast encoding and decoding operations [4]. XtremeDELTA uses more memory to encode, but it achieves the best results while using less memory to decode and apply a delta patch, and it works as fast as SecureDELTA.

Among other binary differencing engines [5], there is an open-source solution provided by xDelta.org, although it seems to be part of an abandoned project today [6]. It employs the vcdiff format in RFC (3284) [7], briefly described in Section 2. The last project entry for the engine is dated 23 April 2016.

In the automotive industry, software-updating applications are compounded in two parts: a server-based system, dealing with creating and deploying updates, and client-side software, factory-installed on each vehicle’s hardware and performing secure update installations.

In this article, we propose a method related to both the server-side software that is designed to create a binary delta file and push it, typically, through over-the-air (OTA) channels to all vehicles that need an update [8]. The method also relates to the client-side software that eventually performs a secure update installation on all required vehicles or computer stations requiring the update.

Our method is an innovative delta-encoding algorithm that embodies data compression as well. We call it Keops. In contrast with other delta algorithms, Keops creates binary-differencing fileswith the main advantage that it outputs a compressed stream even from an initial phase. This is achieved by using a compression distance for comparing the files [9].

This article presents the method and some underlying strategies that help one achieve better performance in terms of speed, compression ratio, and a convenient combination of both. Section 2 presents some existing solutions for delta encoding and relative compression; in Section 3, we cover Keops strategies related to speed and compression improvement. In Section 4, we give some experimental results obtained for automotive big data and present a comparison with publicly available solutions.

## 2. Related Work

In recent years, different industries began using delta-encoding techniques on a large scale, from genome-information data storing, indexing and retrieving to source-code repositories for the automotive, gaming industries or even executable compression [10]. Other examples are project artifacts and source code repositories like Github (a provider of Internet hosting for software development and version control using Git), which display, store, and process source code directly while allowing users to interact with each other and write code remotely in a shared environment. The repositories apply specific algorithms and scripts [11] that allow storing, merging and finding differences within source code [12] while allowing users to better understand code modifications and communicate them among themselves. This helps developers keep track of changes that are performed on source code during different activities, whether they are code formatting or editing, writing new code or just maintaining original code and project artifacts [13].

Most of the references that relate to this subject either implement a modified version of the diff algorithm [7] or use some sort of modified tree [14,15] or graph data structure, which is then noted under different names, such as the abstract syntax tree (AST) and semantic graphs [16].

Falleri et al. [17] introduced an algorithm to compute-edit scripts to tackle abstract syntax-tree granularity, which was modified to include move actions. Their objective was to compute-edit scripts that are shorter and closer to the original intent of a developer. The proposed algorithm was based on analyzing two ASTs, the former made from original source code and the latter made from a modified version of the source code [15]. It is composed of two successive phases: a greedy top-down algorithm that finds isomorphic sub-trees of decreasing height and a bottom-up algorithm in which two nodes match if their children nodes include a large number of common anchors. A search for additional mappings is conducted only when two initial nodes match.

Gerardo et al. [18] proposed an approach that combines vector space models with Levenshtein edit distance to determine if source-code repository differences are due to line additions, deletions, or modifications. It is mostly based on the work of Zimmermann et al. [19] and Ying et al. [20], who computed the differences between classes and methods by matching their names for the purpose of identifying fault-prone modules. Zimmermann et al. [19] introduced the notion of the annotation graph, which is a data structure that represents the line-by-line evolution of a software project over time. The authors’ approach in the present paper improves such a data structure by handling changed lines.

Later, Zimmermann et al. [19] proposed an automated tool called a difference extractor (Dex), which analyzes syntactic and semantic changes in large source-code repositories. This is then applied to source-code repository patches, each of which comprises code changes made to accomplish a particular task. The Dex tool produces summary statistics characterizing these changes for all of the patches that are analyzed. Dex applies a graph-differencing algorithm to abstract semantic graphs (ASGs) representing each version. These differences are then analyzed, and higher-level program changes are then identified.

A binary code update based on binary differencing is also used in the automotive industry. Nowadays, there are hundreds of small electronic control units (ECUs) and controller devices located in modern vehicles; they are mostly responsible for controlling specific functions); thus, updating and maintaining them so they are up to date is a serious task. Once a vehicle leaves its manufacturer, software updates related to bug fixes, security, or any feature update are required to keep the ECUs up and running and updated with the latest software [21]. In Borui et al. [22], the authors discussed a differencing algorithm for reprogramming resource-constrained IoT devices. Called S2, this algorithm was said to be used to achieve small memory and flash footprints by leveraging a topological sort based on an in-place reconstruction mechanism and a stream reconstruction technique; it also achieved a smaller delta size by using prediction-based encoding. The algorithm is based on the bsdiff [7] algorithm, which is known to be very slow at generating patches and only suitable for working with small data buffers. Furthermore, it is well known that time increases linearly by the size of the data buffer while the buffer is processed by using bsdiff.

Westerberg [8] proposed an alternative algorithm based on the update engine from Android. This standalone version of the Android A/B update was implemented and compared with bsdiff [7]. The notion of A/B updates comes from the fact that most ECUs have two identical partitions so that one can be updated while the other is running. The goal is to apply an update as seamlessly as possible. Without a second partition, a car would have to be turned off during the update process. With two partitions, any ECU just requires a restart to be able to switch to the newly updated partition. This is also a ‘backup’ for if the update process goes wrong, allowing a switch back to the old code. The A/B update utilizes different compression methods to generate a patch that is as small as possible. Simplified, it looks at each file in the image and decides what compression method generates the smallest compressed version of that file. An operation representing that compression method is generated for that file [23]. The operations and the data needed to transfer the older version to the newer version are added to a patch file. Once all the operations have been generated, the patch file is sent to a client, where it is applied to the old image in order. That way, a new image identical to the requested version is generated.

In the genome area, a plethora of papers addressed the problem of DNA -information storage and retrieval using relative compression [24,25]. Many data -compression algorithms proposed and implemented within the last two decades, not based on the referenced compression schemes, seemed to perform well but on relatively small sets, such as mitochondrial DNA [26,27]. This pushed them close to impracticality when large sets, such as the human genome [28,29], were in works [30]. The year 2009 saw a rise of referenced-based compression schemes in bioinformatics [31,32,33], with the public release of DZA ZIP [34]. Brandon et al. (2009) added some modifications, and Pavlichin et al. [35] further improved these compression schemes by adding some integer-mapping distribution parameters [36]. As was noted by Ochoa et al., in a genome analysis [37], these algorithms pose some limitations when it comes to a reference-based string, which may need to belong to a specific database available for the human genome only. They further assume that the mapping from the target to the reference is also given.

In 2010, Kurupy et al. [38] proposed a relative LZ (RLZ) compression of genomes for a reference-based compression of an entire set of genomes. Subsequent improvements of this compression scheme, released in the next year, addressed only a modification of greedy parsing into an optimized parsing scheme, which yielded a better compression at the expense of a slower operating time.

Another compression scheme was released in 2011 by Deorowicz and Grabosky [24]; it was called a genome differential compressor (GDC). Surprisingly, this was based on the previously released method RLZ opt [39]. GDC replaced the suffix-tree sub-string search with a hashing-based search, which in the end outperformed RLZ opt, but this was in terms of only compression ratio.

Later, two proposed algorithms, GRS and GReEN, were considered to be state-of-the-art when it came to reference-based genomic data compression. The authors were Wang and Zhang (2011) [40,41], respectively. Few differences arose from the previous work presented. GRS and GRSeEN used only one genome as a reference instead of a set of genomes, and they also assumed a reference was available; thus, no storage of one was required. While GRS considers only pairs of targets and a reference genome sequence, GRSeEN is based on arithmetic coding instead of Golomb encoding. This makes GRSeEN superior to GRS and to the non-optimized RLZ version. However, this was observed only with small sets of genomes, such as bacteria and yeast.

FRESCO is another algorithm that was proposed in 2013; it was designed for the same purpose: to compress a collection of human and non-human genomes. Designed by Wandelt and Ulf (2013) [42], as an innovation, FRESCO allows frequent rewrites of a reference section and embodies a second-order entropy coder.

Part of the reference-based compression schemes is the iDoComp algorithm [37]. It embodies ideas proposed by multiple authors, as described in Christley et al. [34], Brandon et al. [43] and Chern et al. [44]. iDoComp compresses individual genome sequences, assuming a reference is available for both the compressor and the decompressor. The creators chose to use a suffix array to parse the target into a reference due to its attractive memory requirements, especially when compared to other structures, such as the suffix tree. However, the algorithm uses 2 GB of RAM to parse an entire human genome, which makes it less appealing for multi-disciplinary or generic usage.

After carefully reviewing the most important articles, solutions and ideas, we found that the majority of the publicly available solutions do not address the fact that regardless of how good the internally deployed delta algorithm is, there will always be some sort of redundancy the delta algorithm is not addressing. This is simply because by design, a delta algorithm is not a data-compression algorithm. It is a de-duplication algorithm at the best of its abilities.

This is where the novelty of our method comes in. Unlike the vcdiff [7] or ldiff [45,46], bsdiff [7] and the other methods presented in this section, Keops can output already-compressed differencing data streams, even from an initial phase, while performing extremely quickly in a decompression phase. Considering the solutions presented above, xDelta3 [6] can also output already-compressed output streams but does not allow users the possibility of choosing different operating methods to adapt to the ever-changing nature of the input data, which is either highly redundant or less compressible. That is exactly what Keops is able to do. By implementing three strategies, Keops allows users to apply different methods to different types of data by tuning an algorithm depending on whether or not the data are highly compressible.

## 3. Keops Algorithm

Our innovative Keops algorithm derives from the ubiquitous LZ77 [47] data- compression algorithm and uses it as a preferred compression method internally. LZ77 achieves compression by splitting a stream of data to be compressed, also known as an input stream, into two portions; the data are divided by using a current processing pointer called the current or compression pointer *cp*. The two sections are called LZ77 History and LZ77 Look Ahead, respectively (Figure 1). LZ77 History represents past data that has already been processed and compressed up to the current pointer. Obviously, past the current point (including the *cp*) are data yet to be processed, which creates the LZ77 Look Ahead section—hence the name.

LZ77 achieves compression by replacing a common sub-string in the Look Ahead area with pointers for the exact same data and length from the LZ77 History part. The search mechanism employs various data structures and algorithms, such as hash tables, binary search trees and PATRICIA trees [48]. Several heuristic methods are implemented as well, among which, it is worth mentioning, are the greedy and lazy heuristic methods [49]. However, the optimal search strategy [50] is an entirely new domain that we do not cover in this paper.

LZ77, in its original form until the 1982 variation from Store and Szymansky [51], encoded a sub-string or matched text into a triplet <D,L,c> denoting the distance, the length and the uncompressed symbol *c* following this match.

With respect to binary differencing methods, there is one worth mentioning: the *vcdiff* [52], described in IETF’s RFC 3284 [7]. Just like LZ77, this algorithm replaces common sub-strings between two files but encodes them in a series of commands:ADD: Specifies a sequence of bytes to be copied.COPY: Specifies a sub-string that must be copied from source to target.RUN: Specifies a single symbol that will be copied from source to target.

The algorithm coins the terms ‘source’ and ‘target’ windows. In vcdiff [7,52], ‘source’ refers to an old version of a file and ‘target’ refers to a newer version of that file. Three commands are applied to the ‘source’ window in order to extract the differences from the ‘target’ windows. In short, the algorithm replaces sub-strings found between an old and a new file with the above commands, thus instructing a decoder to reconstruct the content in the newer file based on the old file content. No compression is achieved during this process.

### 3.1. Delta File

Inside Keops, the LZ77 encoder acts the same within the *vcdiff* ‘source’ and ‘target’ windows, but unlike *vcdiff*, it instructs the decoder to reconstruct the content of the ‘target’ using the triplets <D,L,c>. Thus, it achieves compression in the binary delta file. The differences from the original LZ77 structure are that LZ77 History consists entirely of the old version of the file and the LZ77 Look Ahead is entirely made up of the newer version. To process LZ77 History, portions from the old file are moved into the LZ77 History location and portions of the new version are moved into the LZ77 Look Ahead location (Figure 2). We call these portions buffers or chunks. The files partition into chunks; their pairings are discussed further in the next section.

The delta file created by Keops is a sum of all the LZ77 compression operations applied to the LZ77 Look Ahead buffer—in a finite number of steps and various history-buffer combinations. The way the steps are controlled and how the two buffers are combined will be further explained.

The first step to creating the Keops delta file is starting with a cp initialized with zero and advancing it all the way through the LZ History buffer. This buffer is parsed, but an output is neither recorded nor saved into the final delta file. When step one completes, the cp is set at the end of the LZ History, regardless of the data size within the LZ History buffer. The cp then matches half of the LZ Buffer, and while advancing it, the LZ77 starts parsing the rest of its buffer. While parsing the LZ Look Ahead buffer, the LZ77 output is recorded into the delta file. At the end of the LZ Look Ahead parsing, Keops outputs all the triplet <D,L,c> it can find, which include encoded matches from the LZ Look Ahead to the LZ History and all uncompressed symbols, if any. Eventually, the Keops archive will be a sum of all the output series from all the buffers from the new file, regardless of the strategy employed to pair the buffers. The pairing strategies that make up the Keops modelling techniques will be discussed further in the next section.

### 3.2. Strategies for Buffer Pairing

With LZ77, the more alike the History and Look Ahead buffers are, the better the compression is. In Keops, when source buffers are paired with target buffers, we have the liberty of choosing, as a history, the most similar buffer to the source. This buffer may have the same index as the current target buffer, may be shifted upward or backward because of a block removal or a new code insertion in the target or may simply vanish because the code update was too radical. Any search supposes the calculation of a distance. In the case of Keops, since the main goal is to obtain a delta file as small as possible in size, we choose to use a generalized compression distance. This distance is calculated by applying LZ77 to the source and target buffers as described in the previous section. The size of the compressed target buffer represents the distance between the two buffers.

In the following sections, we present three strategies for pairing the buffers. They are designed to optimize either the compression time or the compression rate or to balance them.

#### 3.2.1. One-to-One Strategy (Time Optimized)

When a set of changes is designed to update or improve a software or its associated data file, it is usually called a patch. Called bug fixes or simply fixes, patches are usually designed to improve the functionality of a program or fix a coding flaw. Design flaws are more complicated and they involve many, if not structural, changes to the software. However, for bug fixes and small changes, most of the time, it is easier and more economical to distribute patches to users rather than to redistribute a newly recompiled or reassembled program. Software patches are pictured in Figure 3.

In the case of patches, the one-to-one strategy is the best solution. Since modifications are small and usually localized within the same code block of the flawed code, there is a higher chance for the change to also be within the same code block. There is also a very high chance that the code block will also maintain the same size. The one-to-one strategy pairs buffers with the same positions in the source and target codes.

These buffers are, with a high probability, rather similar if not identical, which means a high compression of a generated delta file. The other advantage is that no search of a corresponding buffer has be done at the level of the target, its position being known in advance. This significantly increases the processing speed of Keops. The one-to-one strategy is depicted in Figure 4.

#### 3.2.2. Brute-Force Strategy (Rate Optimized)

When differences between the old and the new file are numerous, the one-to-one strategy cannot offer a good compression anymore. Consider the case in Figure 5, where in the new file, data are mostly new, but data from the old file are still present. It should be noted that because of code additions, modifications or substitutions, the one-to-one mapping is no longer preserved, since in this situation, the files are out of sync. This is a classic example of software being redesigned when very little old code will be kept.

The de-synchronization of the code necessitates a search for the most similar buffer in the target. In Keops, the search is conducted by using the LZ77-based compression distance. The brute-force strategy supposes an exhaustive search over the entire source code. This increases the probability of a good match, although it is not optimal as long as the buffers in the target file are processed sequentially. However, our experiments have shown that with this strategy, we are close to the optimal pairing, and consequently, the best compression for this type of updated software. Figure 6 depicts the buffer pairing by using the brute-force strategy. The drawback of this strategy is the longer overall processing time of Keops, but this is the price of preserving a close-to-optimal compression.

#### 3.2.3. Flexi Strategy

Within updated code or data which may count numerous differences, the most similar buffer of the source is usually in the vicinity of the considered target buffer. This is explained by the fact that the de-synchronization is produced by both removing and adding code blocks; thus, the blocks’ shift is not accumulated. An example is in Figure 7. The flexi strategy addresses this aspect by searching in a limited domain around the position of the considered target buffer. This may be a good compromise between the compression and processing speeds. In Figure 7, with the flexi strategy, the search is done up and down with four positions.

## 4. Experimental Results

We tested the Keops binary delta encoder on five types of data. The first test series comprised the minGW compiler binaries for Windows platform (mingw). We created binary delta files between versions 4.4 and 4.5 using various sizes for the History and Look Ahead buffers.

The second type consisted of formatted text files representing software performance logging files, or trace files, combined together into a single file. The files are called ALog and ALog2. The latter represents the same archived files as ALog, minus certain files that have been removed. They contain English-language text among which there is a high frequency of file names, operation dates and times, file sizes, file paths, etc.

For the third test package, we chose two binary images from a collection of ECU binary images specifically used in one of our automotive projects. We chose swfk ver. 20 and swfk ver. 18.

Replay was the fourth test package, which contains a specific media-related Windows software. We used real instances of this software: versions 31.2, 31.4 and 31.5.

For the last test package, we chose silezia corpus files, a well-known collection of English text and binary data files commonly used in data compression tests [53]. We removed file *osdb* from the content of the first corpus, thus creating two corpus instances: silezia 1 and silezia 2.

Three out of five test packages—ALog, swfk and Replay—are specific to the automotive environment.

The test packages were chosen to be representative of various or homogeneous data, low- and high-redundancy files and structured and binary files. The files’ sizes and their roles (targets or sources) are mentioned in Table 1, Table 2 and Table 3. The tests were done for five buffer sizes: 2, 4, 8, 16 and 32 MB. We used equal History and Look-Ahead buffers. All tests were run on a Windows 10 PC running on an Intel i3-4130 CPU at 3.40GHz with 8 GB of RAM.

We tested Keops using the three strategies presented in Section 2: one to one, brute force and flexi. For each of them, we recorded the delta rate, the encoding and decoding time and the memory requirements. The delta rate was the compression rate in percentage, expressed as the ratio of the sizes of the targets after the Keops compressions and before them. The results are presented in Table 1 (one-to-one strategy), Table 2 (brute-force strategy) and Table 3 (flexi strategy). To evaluate the effectiveness of the differencing concept with respect to a simple compression, we compared the delta rate with the ZIP rate calculated as the ratio of the size of the ZIP-compressed target and the size of the uncompressed target.

For the flexi strategy, the search of the most similar blocks in the source was conducted up and down with K positions relative to the current block in the target file, where K had a value of four by default. We chose this range after having observed the distribution of the gap of similar blocks in our experimental data (Figure 8). The distribution tended to be concentrated around zero, disregarding the file type. For Alog and silezia, where the newer versions were obtained by using removals, the distribution was bi-modal with a mode gap of two to six blocks, respectively. For mingw and swfk, which have less redundant versions, the distribution was more dispersed. In their case, limiting the search at four gaps degraded the compression, as can be seen from Figure 9. A small block size also spread the distribution (Figure 10), meaning that it is recommended to increase the search area when such blocks are used.

### 4.1. Compression Rate

We first analyzed the impact of the block size on the compression. The plots in Figure 9 show that generally, the delta rate improves with the block size. This effect tends to be less important only when working with larger blocks, especially the 16 and 32 MiBs ones. There are many exceptions to this rule of thumb because binary delta creation is a very data-dependent process. If more differences are spotted between the old and the new data, the fewer the matches between them are found by Keops. In conclusion, if data have an extremely low redundancy stream comprised of old and new buffers (files, respectively), the less important the block size becomes.

Concerning the strategies, the plots in Figure 9 show that brute force is generally the best if the goal is a good compression. However, there are exceptions, e.g., for the Replay package, for which all strategies gave the same delta rate (the plots were superposed) or for the Alog package, for which brute force and flexi behaved identically.

It should be noted that delta-rate curves are convergent as block sizes increase. Thus, if 32 MiB blocks are used, one should choose the one-to-one strategy, which is time optimized. There is no reason to do any block search and introduce delays as long as the compression is the same.

As the results in Table 2 show, the delta rate of Alog reached 0.15%. This means the delta file was less than 1% in size compared to the target size. It was a very good result in line with our predictions, especially when compared to the ZIP rates of 7.96%. This shows that the Keops algorithm performs extremely well when it is applied to highly redundant data due to content similarities.

For mingw and swfk, however, when having less redundant versions, the minimum delta rates were 41.95% and 14.98%, but they were still lower than the actual ZIP (deflate) compression rates shown in Table 2.

Generally, the difference between the delta rate and ZIP rate grew when large history buffers were used by Keops. For example, Replay shows an improvement in the delta rate from 22.15 to 1.33% as compared to a 75.78% ZIP rate.

We included the ZIP compression rate results of the new version (target file) in the tables as well since many software applications, even today, do not have software-update capabilities at the binary-differencing level. They simply allow users to download (possibly) a new compressed version of their application. There are many cases in which users download a ZIP file with an installer that handles a new version, i.e., uninstalls old versions and installs a newer version of the same software. There is a noticeable difference between the Keops binary delta file size and the ZIP archive, the compressed version of the newer software version. Usually, any compatible ZIP or ZIP64 archiving software works by implementing an LZ sliding window that cannot be larger than 32 KiB or 64 KiB, respectively, in order to not abide by the zipping’s deflate-algorithm RFC specifications [54]. Since only the new file is compressed, there is no immediate history to be used within the LZ algorithm; thus, the LZ table is empty at the start position zero. Even once the compression pointer advances within this ZIP-compatible algorithm, no more than 32 or 64 KiB of previously seen data can be used as a history. This is a complete disadvantage when comparing it with Keops, which starts with fully loaded previously ‘seen’ data (from the old file) and works with a minimum of a 2 MiB buffer as an LZ History. As can be seen from the tables, for our experimental data, the difference between the ZIP rate and the delta rate may vary anywhere from 7 to 75.

### 4.2. Encoding Time

As expected, the encoding time depends on the strategy. Brute force and flexi, which included a search for similar blocks, demanded longer encoding times. On average, brute force had four-times longer durations, while the flexi strategy was only 2.5 more time consuming. Obviously, the highest encoding times correspond to encoding with small blocks.

Another interesting fact about the test results we have seen so far is that with some test files, as the History buffer size grew, so did the encoding time. In some other cases, the encoding time shrank. We can explain this anomaly with the file redundancy. More precisely, the more common sub-strings are found between the old data and the new data, the faster Keops’s LZ77 algorithm performs. It accepts sub-strings longer than a certain threshold, e.g., 2048 bytes, and no other searches are performed. This way, the compression pointer advances more rapidly and the data are consumed much faster in this case. Larger buffers and highly redundant files tend to allow for much longer sub-strings than the ones found with smaller buffers. Smaller buffers tend to fragment larger common sub-strings while requiring an inertial time and bytes to pick up the common sub-string again as new blocks are loaded.

### 4.3. Decoding Time

While the encoding times go from 3.75 s to 1253 s, the decoding is much faster. Depending on block size, file type and strategy, it may have values between 0.18 s and 10.46 s. Moreover, we noticed that usually, it goes lower once the block size increases, except with the 32 MiB blocks. This could be related to the fact that larger buffers tend to allow for more distant matches, and once matches (or common sub-strings) are further away, there is a higher probability cache misses and page faults will occur. More distance data needs to be copied from the matched pointer to the current decompression pointer, increasing the number of page faults. We documented this case. With Replay (31.4 to 31.5), from a block size of 16 MiB to a block size of 32 MiB, the number of page faults jumped from 34,189 to 46,493, a good explanation of the increased decompression time from 0.203 to 0.766 ms. This can be corrected with a slight loss of compression by abandoning smaller but distant matches in the compression step. The zlib library [54] performs that for matches that equal a certain threshold and distance surpassing 4 KiB.

By design, Keops is built on the idea of maximizing decompression speed. The modified LZ engine is parameterized for maximum sub-string searches. Every sub-string found using this engine has the potential to speed up the decoding process. To find most of the available sub-strings between the old file data and the new file data, exhaustive searches are performed. Hash tables are designed for a maximum occupancy, and hash collisions are searched without a set limit. Keops also employs smart heuristics for matches. Some heuristics are meant to minimize page faults, which, in turn, maximize speed.

### 4.4. Memory Requirements

Being an LZ-powered algorithm, Keops behaves as a memory hybrid, requiring less memory to reconstruct the compressed data during the decoding step then it requires to actually compress the data. This is due to the fact that no searches are performed during the decompression step. None of the search structures need to be allocated, nor do they consume any memory except during the encoding step.

Looking at Table 1, Table 2 and Table 3, it is clear the memory requirements are almost the same for all three Keop variants, e.g., the mingw test file sets, for which the encoding process required 325 MiB of RAM while the decoding process required 273 MiB of RAM for all three Keops variants. The difference in memory occupancy was exactly the search structure sizes that were missing and were not required to be allocated and initialized in the RAM during the decoding process.

Between the encoding and decoding processes, there is another memory-related difference, which consists of the fact that during the encoding process, memory is initially occupied with the exact sizes of both the files, old and new, right after they are processed in blocks as described in this article. During the decoding process, the memory is initially filled with only the old file blocks, followed by the new file, which is built block by block depending on the chosen strategy. Of course, depending on the chosen strategy, some partial writing (dumping) into the output file can be performed, further minimizing the impact of RAM usage during the decoding step.

By relying less on memory caching and by exhaustively using the storage devices where source and target files are stored, the memory requirement for each strategy can be significantly reduced to as low as twice the minimum block size (e.g., 4 MiB of RAM for the one-to-one strategy when using a 2 MB LZ window size) for both the encoding and decoding processes.

In conclusion, the described Keops’ memory requirement is purely an implementation decision. Changing it does not affect the end results in any way, shape or form when it comes to the final compression rates or the final update package sizes. It may only affect the processing time, since disk and memory trashing may occur due to the exhaustive reads of file blocks and the continuous dumping of memory buffers after a single block has been processed.

The authors of this algorithm were more interested in these final compression results and update sizes rather then turning this idea into a commercially viable implementation, which can be achieved with a bit more attention to this single factor—the used memory.

### 4.5. Comparison with Existing Solutions

We compare Keops with three publicly available binary-differencing file creators, SecureDELTA [2], XtremeDELTA [3] and xDelta3 [6], the first two being commercial and closed-source software and the last one being open source. The compression rate and the encoding/decoding times are given in the Table 4, Table 5 and Table 6.

A first remark concerns the encoding times, which are systematically longer than those of the three differencing engines. The Keops encoding times range between 3.5 s and 488.27 s depending on the file, while for the considered engines, it is no longer than 69 s. Keops gains at the level of the decoding time, which is considerably shorter, except for with Replay, for which it is a bit longer but remains comparable. Keops is able to outperform xDelta3 in compression rates in two cases by more than 3% while being two to five times faster in decompression. It is a known fact that [55] is extremely slow in decompression, so choosing this algorithm in this case is an unfortunate design issue.

In comparison with [2] and XtremeDELTA [3], Keops outperforms them in delta sizes because at these current versions, they do not compress the output data in any form. These advanced delta creators, i.e., SecureDELTA [2] and, in particular, XtremeDELTA [3], are designed specifically for embedded software. By design, they use the lowest possible memory when recreating the original files on the target machines, ranging from 2 MiB to 6 MiB of RAM regardless of the input file sizes. This explains why Keops is able to surpass them in decompression speed. For a fair comparison, since SecureDELTA and XtremeDELTA output uncompressed delta files, we calculate, for these two differencing engines, a compressed delta rate as a ratio of ZIP-compressed delta file to Keops-compressed file. Even after compression, Keops remains better than SecureDELTA (except for with the Replay data). XtremeDELTA [3] outperforms Keops for two out five test data packages. xDelta3, as it relies on LZMA [55] as a main algorithm, also produces compressed output files. However, Keops is able to beat xDelta3’s LZMA algorithm and achieves a better compression rate for only the swfk data.

## 5. Conclusions

We designed Keops for environments in which the encoding time for creating a software update is not considered to be an issue. Applying a software update and decoding a binary-level delta package has always been a problem due to different environments with the different resources and configurations the update is supposed to work on. When it comes to decoding, Keops offers a low decoding time, making it more suitable for situations in which decoding time is a priority.

Keops is the best fit for low- to mid-memory environments that need to operate software updates at high speeds, since we provide, depending on the chosen strategy, the best possible solution for compression ratios and, ultimately, binary delta sizes. A more sophisticated software implementation of our Keops algorithm could further optimize the used-RAM footprint with different techniques, such as lowering cache levels and frequently writing blocks into output files depending on the chosen strategy, thus minimizing the time needed for the encoding process.

## Figures and Tables

**Figure 1 entropy-24-00574-f001:**
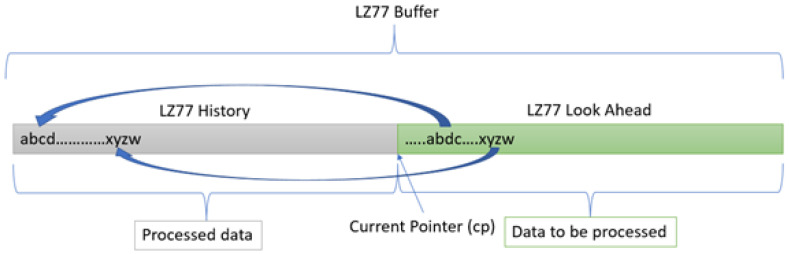
LZ77 and the two sections that form the LZ77 buffer together.

**Figure 2 entropy-24-00574-f002:**
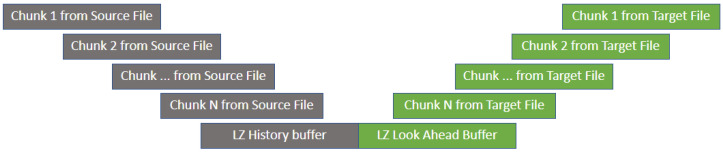
The LZ77 algorithm inside Keops for delta differencing.

**Figure 3 entropy-24-00574-f003:**
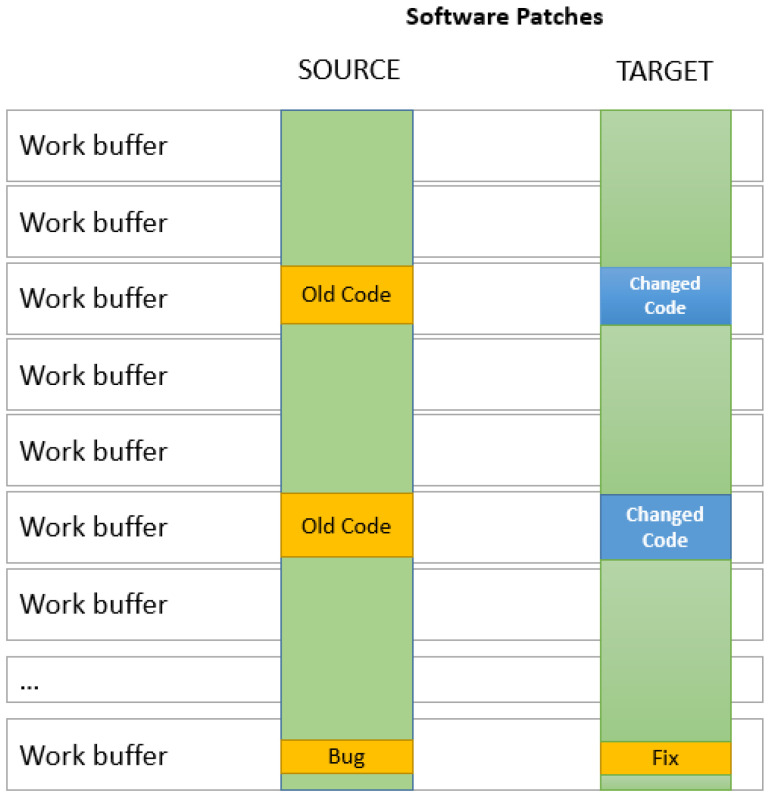
Two file versions before and after software patches are applied. The patches are in yellow.

**Figure 4 entropy-24-00574-f004:**
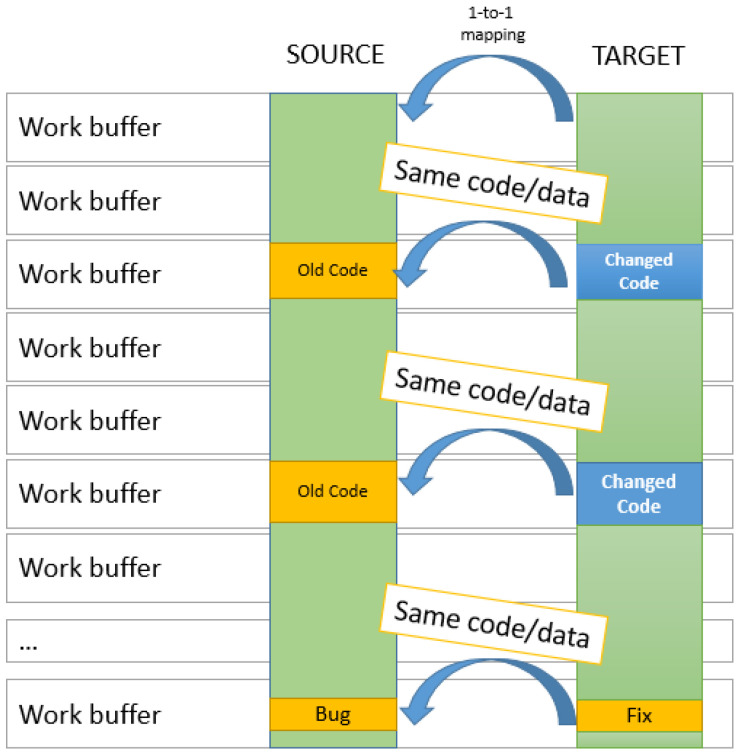
Buffer pairing in one-to-one strategy, best suited for software patching and time optimized.

**Figure 5 entropy-24-00574-f005:**
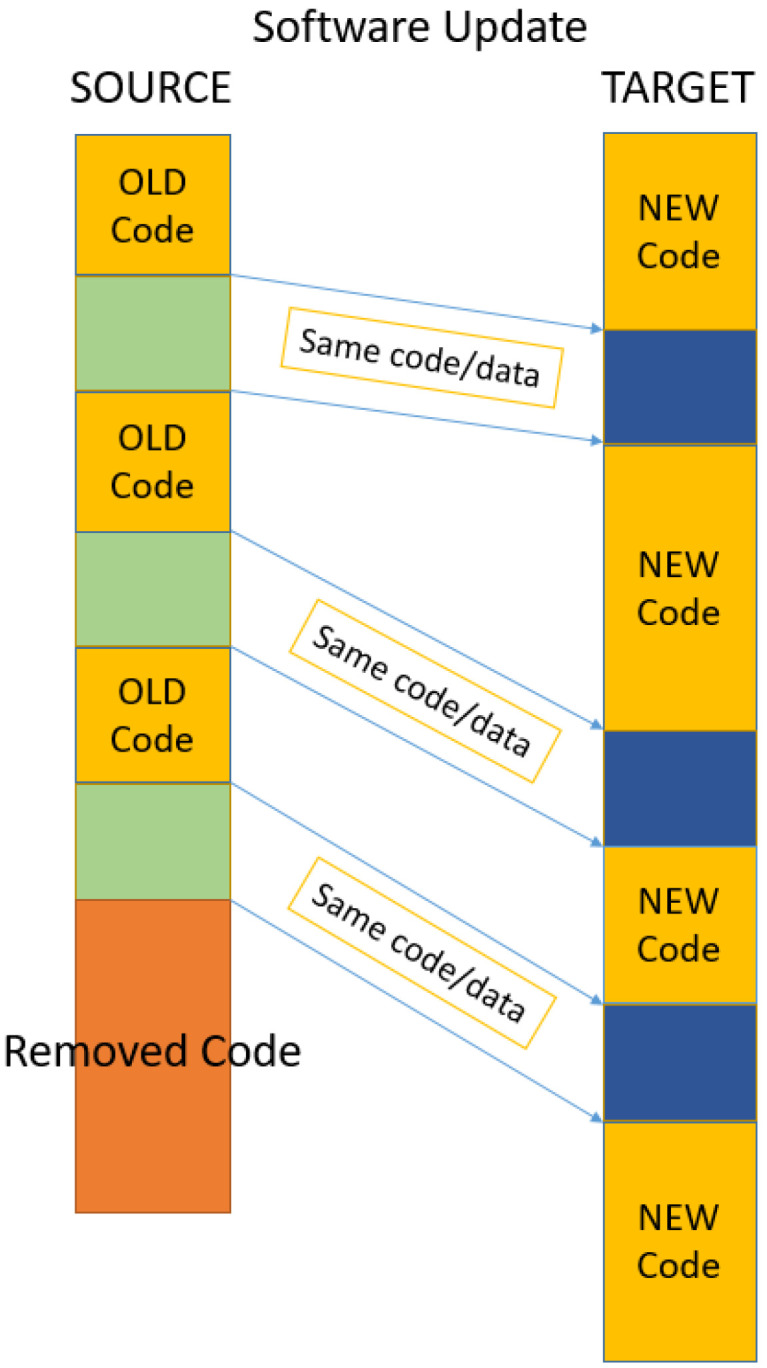
A typical software-update scenario with interleaved matching data at byte level.

**Figure 6 entropy-24-00574-f006:**
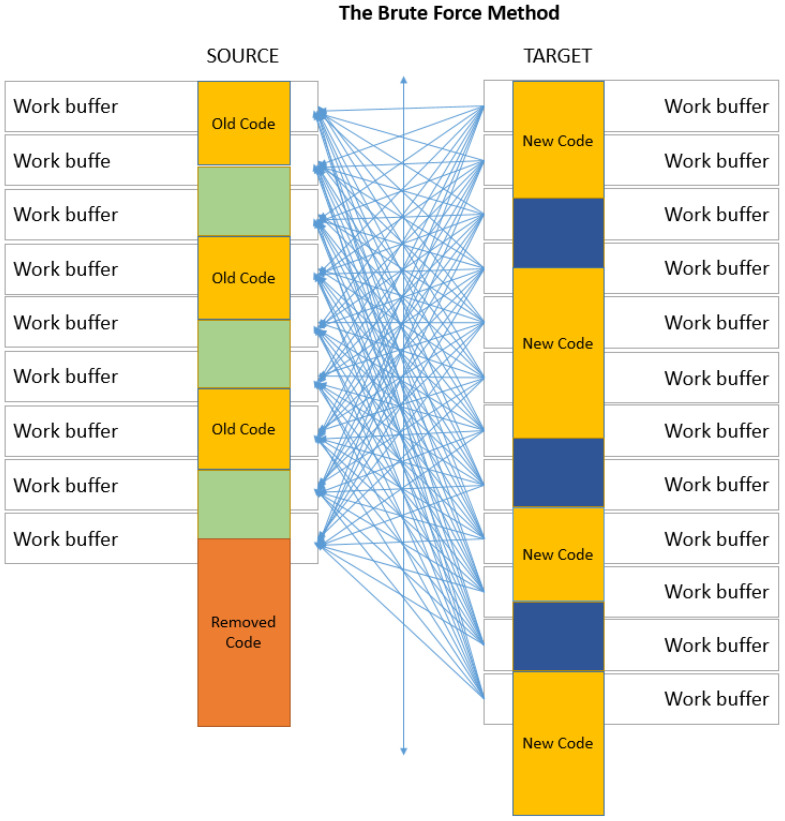
Brute-force strategy when applied to a software update with structural modifications.

**Figure 7 entropy-24-00574-f007:**
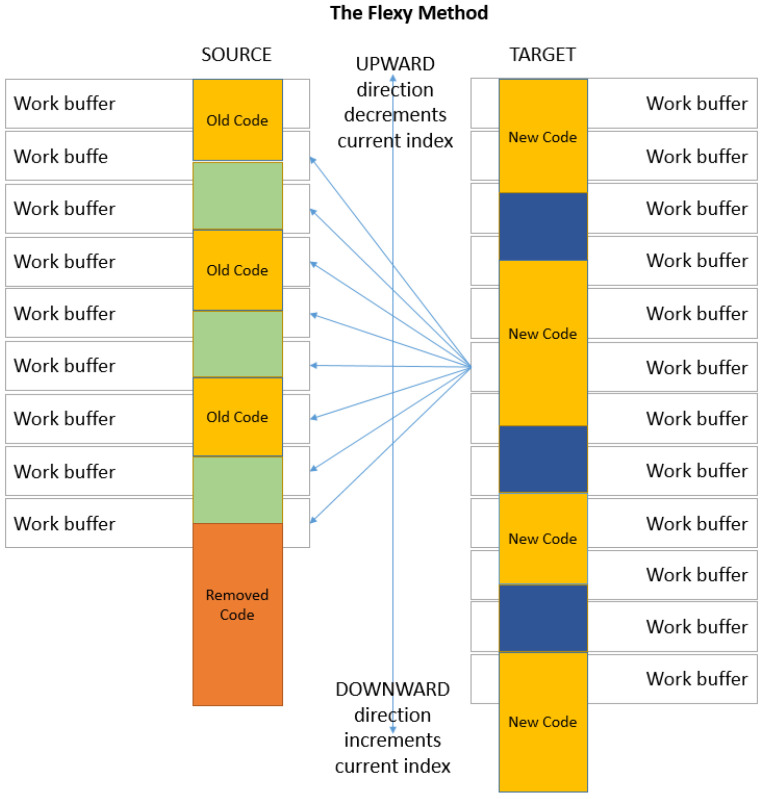
Flexi strategy for search domain limited to four indexes upward and downward.

**Figure 8 entropy-24-00574-f008:**
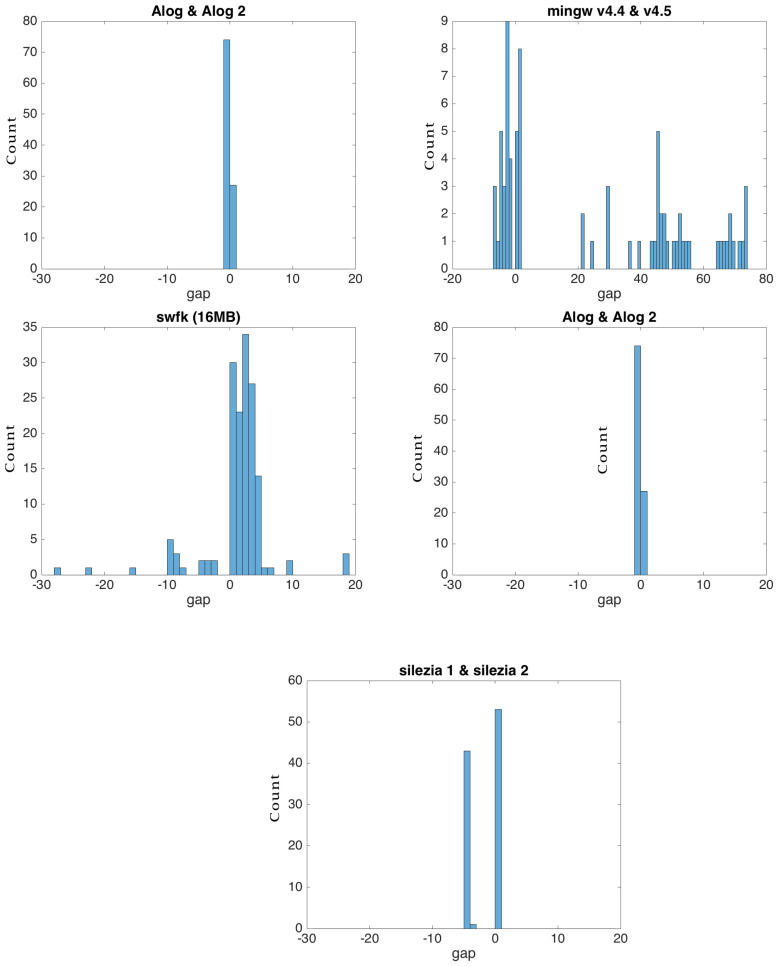
Distribution of similar block gaps in the source and target files for the experimental data and 2 MB block size buffers.

**Figure 9 entropy-24-00574-f009:**
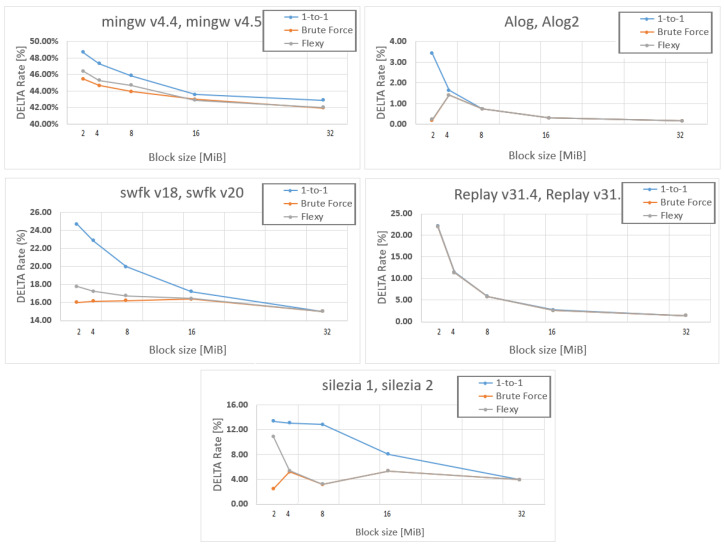
Delta rates vs. block sizes.

**Figure 10 entropy-24-00574-f010:**
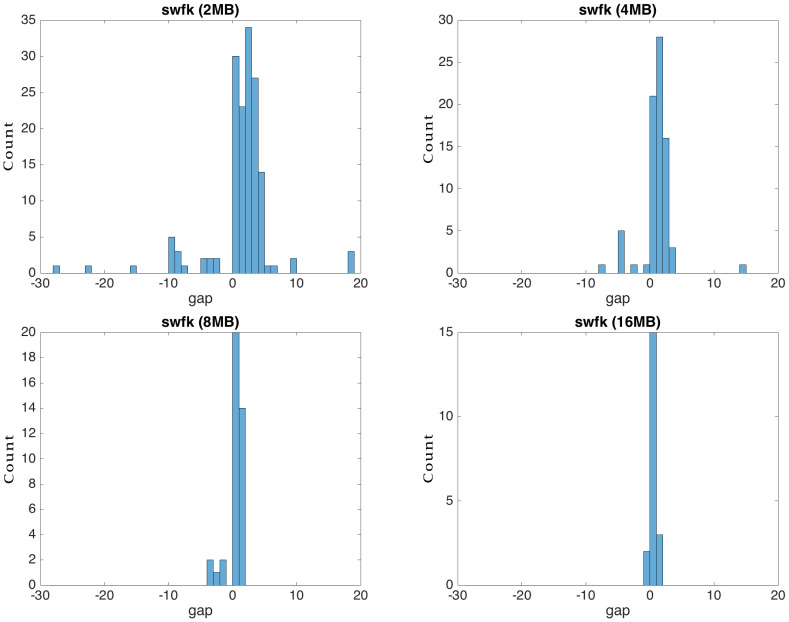
Gap distribution of swfk for various block sizes. The distribution stretching indicates that search range of flexi should be adapted to the block size in order to keep the encoding time short while maintaining a good compression.

**Table 1 entropy-24-00574-t001:** Results for one-to-one strategy (time optimized).

Source [Bytes]	Target [Bytes]	History Buffer [MBs]	Encoding		Delta Rate [%]	Decoding		ZIP Rate [%]
			Time [s]	RAM [MiBs]		Time [s]	RAM [MiBs]	
mingw ver. 4.4	mingw ver. 4.5	2	85.671	325	48.67	3.843	273	52.77
116.858.661	159.723.368	4	125.657	363	47.28	3.766	277	
		8	182.000	439	45.85	3.375	286	
		16	232.203	583	43.56	2.85	302	
		32	312.078	887	42.88	5.265	350	
Alog	Alog2	2	75.546	466	3.42	0.688	313	7.96
213.403.728	211.206.693	4	58.813	503	1.64	0.703	317	
		8	35.156	583	0.73	0.812	429	
		16	18.812	727	0.31	0.89	444	
		32	13.047	1047	0.15	0.453	396	
swfk ver. 18	swfk ver. 20	2	251.547	692	24.72	10.406	640	30.42
340.375.413	319.794.721	4	323.000	732	22.86	9.594	646	
		8	372.625	807	19.97	8.469	654	
		16	427.843	967	17.18	7.406	678	
		32	488.266	1271	14.98	10.344	726	
Replay ver. 31.4	Replay ver. 31.5	2	5.593	159	22.15	0. 360	107	75.78
50.941.859	51.583.426	4	5.016	199	11.44	0.250	113	
		8	4.860	279	5.83	0.234	125	
		16	4.300	439	2.67	0.187	149	
		32	3.750	727	1.33	0.704	181	
silezia 1	silezia 2	2	97.047	458	13.36	4.125	405	41.75
211.961.968	201.876.111	4	146.563	495	13.05	3.437	409	
		8	204.625	575	12.81	3.313	422	
		16	178.194	727	8.04	2.453	438	
		32	101.359	1047	3.94	2.141	422	

**Table 2 entropy-24-00574-t002:** Results for brute-force strategy (rate optimized).

Source [Bytes]	Target [Bytes]	History Buffer [MBs]	Encoding		Delta Rate [%]	Decoding		ZIP Rate [%]
			Time [s]	RAM [MiBs]		Time [s]	RAM [MiBs]	
mingw ver. 4.4	mingw ver. 4.5	2	520.922	323	45.46	3.125	273	52.77
116.858.661	159.723.368	4	408.547	363	44.66	3	277	
		8	356.406	439	43.93	3	285	
		16	330.906	578	42.99	2.766	301	
		32	406.734	887	41.95	4.344	349	
Alog	Alog2	2	158.515	466	0.19	0.875	413	7.96
213.403.728	211.206.693	4	120.407	503	1.41	1.141	417	
		8	70.219	583	0.73	0.765	429	
		16	36.625	727	0.31	0.671	444	
		32	23.64	1047	0.15	0.438	396	
swfk ver. 18	swfk ver. 20	2	2253.625	690	15.98	9.593	640	30.42
340.375.413	319.794.721	4	1253.062	727	16.12	7.719	646	
		8	890.218	807	16.2	7.844	654	
		16	744.282	962	16.34	7.297	678	
		32	633.297	1271	14.98	11.156	726	
Replay ver. 31.4	Replay ver. 31.5	2	111.687	159	22.05	0.031	56	75.78
50.941.859	51.583.426	4	81.312	199	11.33	0.25	113	
		8	49.547	279	5.72	0.235	125	
		16	27.765	439	2.56	0.188	149	
		32	13.688	727	1.33	0.797	181	
silezia 1	silezia 2	2	421.797	458	2.38	1.094	405	41.75
211.961.968	201.876.111	4	324.469	495	5.27	1.625	409	
		8	187.344	575	3.14	1.234	421	
		16	175.687	727	5.33	1.61	438	
		32	143.203	1047	3.94	4.907	422	

**Table 3 entropy-24-00574-t003:** Results for flexi strategy.

Source [Bytes]	Target [Bytes]	History Buffer [MBs]	Encoding		Delta Rate [%]	Decoding		ZIP Rate [%]
			Time [s]	RAM [MiBs]		Time [s]	RAM [MiBs]	
mingw ver. 4.4	mingw ver. 4.5	2	152.98	325	46.38	3.66	273	52.77
116.858.661	159.723.368	4	201.36	363	45.28	3.20	277	
		8	270.00	439	44.7	2.94	285	
		16	358.97	583	42.89	2.63	301	
		32	453.77	887	42.03	4.75	349	
Alog	Alog2	2	22.53	466	0.23	0.81	413	7.96
213.403.728	211.206.693	4	52.25	503	1.41	1.38	417	
		8	44.14	583	0.73	1.11	429	
		16	30.53	727	0.31	0.66	444	
		32	22.36	1047	0.15	0.45	396	
swfk ver. 18	swfk ver. 20	2	254.88	692	17.75	7.22	640	30.42
340.375.413	319.794.721	4	337.81	732	17.22	8.41	646	
		8	446.17	807	16.73	7.38	654	
		16	514.02	967	16.41	7.11	678	
		32	561.39	1271	14.98	10.83	726	
Replay ver. 31.4	Replay ver. 31.5	2	39.94	159	22.05	0.33	107	75.78
50.941.859	51.583.426	4	51.03	199	11.33	0.24	113	
		8	49.72	279	5.72	0.28	125	
		16	31.00	439	2.56	0.19	149	
		32	15.28	727	1.33	0.66	181	
silezia 1	silezia 2	2	155.83	458	10.89	4.13	405	41.75
211.961.968	201.876.111	4	142.24	495	5.36	3.44	409	
		8	116.80	575	3.19	3.31	422	
		16	165.92	722	5.33	2.45	438	
		32	151.92	1047	3.94	2.14	422	

**Table 4 entropy-24-00574-t004:** Comparison of Keops with publicly available binary difference engine SecureDELTA.

	SecureDELTA				Keops		
	Encoding Rime [s]	Delta Rate [%]	Compressed Delta Rate [%]	Decoding Time [s]	Encoding Time [s]	Delta Rate [%]	Decoding Time [s]
mingw	17	68.09	48.1	10	312.08	42.88	5.265
Alog	2	0.0035	0.0035	10	13.05	0.15	0.453
swfk	45	36.81	15.85	43	488.27	14.98	10.344
Replay	5	0.62	0.39	1	3.75	1.33	0.703
Silezia	18	0.00242	0.0024	52	101.36	3.94	2.141

**Table 5 entropy-24-00574-t005:** Comparison of Keops with publicly available binary difference engine XtremeDELTA.

	XtremeDELTA				Keops (One-to-One Strategy, 32 MB Blocks)		
	Encoding Time [s]	Delta Rate [%]	Compressed Delta Rate [%]	Decoding Time [s]	Encoding Time [s]	Delta Rate [%]	Decoding Time [s]
mingw	23	37.07	25.48	9	312.08	42.88	5.265
Alog	3	0.001	0.001	12	13.05	0.15	0.453
swfk	69	35.32	15.88	46	488.27	14.98	10.344
Replay	2	0.59	0.39	1	3.75	1.33	0.703
silezia	10	0.001	0.001	46	101.36	3.94	2.141

**Table 6 entropy-24-00574-t006:** Comparison of Keops data with publicly available LZMA compression-based binary difference engine xDelta3.

	xDelta3			Keops (One-to-One Strategy, 32 MB Blocks)		
	Encoding Time [s]	Delta Rate [%]	Decoding Time [s]	Encoding Time [s]	Delta Rate [%]	Decoding Time [s]
mingw	42	45.9700	11.31	312.08	42.88	5.265
Alog	14	0.0007	1	13.05	0.15	0.453
swfk	63	16.2000	23	488.27	14.98	10.344
Replay	1	0.2700	1	3.75	1.33	0.703
silezia	15	0.0028	14	101.36	3.94	2.141

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
