# Peer review of "A Hybrid Data-Differencing and Compression Algorithm for the Automotive Industry"

_entropy, 2022, doi:10.3390/e24050574_

Round 1

Reviewer 1 Report

Please see the attached file for comments.

Author Response

Comments to the Author

Comments for ”A Hybrid Data Differencing and Compression Algorithm for the Automotive Industry”(entropy-1645238) This manuscript presents a method to measure (estimate) the step length by using 3D LiDAR sensors. The motivation of this work is interesting and exhibits great novelty. The early detection mechanism for people aging is important. However, how the step length relates people aging is not clearly stated.

Response: Thank you for your time in reviewing and writing comments about our article. However, we do not understand the connection between the 3D LiDAR and our Hybrid Data Differencing method.

Reviewer Comment:

In Abstract, the comparison results should briefly be summarised. Also, some important keywords should appear in the Abstract as well. For example, the terms “OTA software update” and “LZ77” are not shown in the Abstract.

Response: We have added the missing terms to the abstract and we briefly summarized the most outstanding results. The abstract now states:

"We propose an innovative delta differencing algorithm that combines software updating methods with LZ77 data compression. This software updating method relates to the server-side software that creates the binary delta file and also to the client-side software that performs the software update installation. The proposed algorithm creates binary differencing streams already compressed from the initial phase. We present the software updating method suitable for OTA software update, and its basic strategies to achieve better performance in terms of speed, compression ratio, or a combination of both. A comparison with publicly available solutions is provided. Our test results show Keops can outperform an LZMA based binary differencing solution on compression ratio in two cases with more than 3%, while being 2 to 5 times faster in decompression. We also prove experimentally that the difference between Keops and other competing delta creator software grows even bigger when larger history buffers are used. In one case we achieve a 3 times better performance in delta rate as compared to other competing delta rates."

Reviewer Comment:

In Sec. 1, the full name of the abbr. “LZ” should be given. Moreover, the corresponding reference is required.

Response: We have given many references and description of acronyms related to Lempel-Ziv dictionary based compression and also cited the appropriate articles and authors of such methods and algorithms. However, in this case we solved the comment by changing the text to "a fully fledged dictionary based compressor." (line 23)

Reviewer Comment:

Near line 51, the authors stated “ Technologies srl a...”. What is srl?

Response: We are supposed to write the exact name of the trademark owner or software proprietor which owns the rights of such programs or software product names. We have added a footnote for SRL/srl to state what it means i.e., Limited Liability Company for Romanian companies. (pg. 2)

Reviewer Comment:

In Sec. 2, the research gap should be pointed out clearly. The reviewer suggests adding one more paragraph at the end of Sec. 2 to point out the gaps.

Response: We have updated Section 2 by adding at the end two paragraphs pointing out the weaknesses of presented methods, what is the gap we are trying to fill in and how are we addressing these issues.

After carefully reviewing the most important articles, solutions or ideas, we find that the majority of the publicly available solutions do not address the fact that regardless of how good the internally deployed delta algorithm is, there will always be some sort of redundancy the delta algorithm is not addressing. This is simply because by design, a delta algorithm is not a data compression algorithm. It is a deduplication algorithm to the best of its abilities.

This is where the novelty of our method comes into play. Unlike the vcdiff, bsdiff, or any other method we have presented in this section, Keops can output already compressed differencing data streams, even from the initial phase, while performing extremely fast by design, in the decompression phase. From the above presented solutions, xDelta3 can also output already compressed output streams but does not allow the users the possibility to choose different operating methods to adapt to the ever-changing nature of the input data, either highly redundant or less compressible. It is exactly what Keops is able to address. By implementing three strategies, Keops allows the users to apply different methods to different types of data by tuning the algorithm depending on whether the data is highly compressible or not.”

Reviewer Comment:

How do the authors come up with the name “Keops”? The novelty of Keops algorithm is unclear.

Response: Keops was a personal choice. It came as a simple association between Big data and a huge pharaonic monument. Another option we have considered was HDCA (Hybrid data Differencing and Compression Algorithm) but it seemed us difficult to memorize.

The novelty consists in the fact that unlike some standards like vcdiff, bsdiff or any other method we have presented in Section 2, Keops is able to output an already compressed differencing data stream from the initial phase, while performing extremely fast by design, in the decompression phase. We have clearly stated this in the paper by adding in the end of Section 2 the above mentioned two paragraphs.

Reviewer Comment:

The labels of y-axis are not suitable. I think it should be “count” rather than frequency.

Response: Figures have been modified to state “count” instead of “frequency”

Reviewer Comment:

The mathematical definition of the delta rate and zip rate is required.

Response: We defined DELTA rate and ZIP rate in the following paragraphs added at lines 359 and 362.

“DELTA rate is the compression rate in percentage, expressed as the ratio between the sizes of the target after Keops compression and before it.

….

To evaluate the effectiveness of the differencing concept by respect to a simple compression, we compare DELTA rate with ZIP rate, calculated as the ratio between the size of ZIP compressed target and the size of uncompressed target.”

Reviewer Comment:

In Table 4, what is the compressed delta ratio?

Response: We defined compressed DELTA rate in the following paragraphs added at line 511.

“For a fair comparison, since SecureDELTA and XtremeDELTA output uncompressed delta files, we calculate for these two differencing engines also a Compressed DELTA Rate as the ration between ZIP compressed delta file and Keops compressed file.”

To see the updated version of the paper, please download the attached file.

Reviewer 2 Report

Some comments are list below:

  1. Regarding the proposed hasting algorithm, lacking descriptions on contribution manner and its functionality obligated extending distinguish.
  2. The author should provide further descriptions of the paper contributions and detailed reason why the proposed approach enhances existed works.
  3. The general definitions of index terms should be lessened to strengthen paper to meet technical sound.
  4. Acronym, the authors haven’t written the full meaning of word (ECUs) at line 126
  5. Acronym, the authors haven’t written the full meaning of word (LZ) at line 165
  6. Acronym, the authors haven’t written the full meaning of word (RLZ) at line 172
  7. The authors haven’t forgotten remove empty of two pages at the last of Journal Paper
  8. In the Reference section, the link of title is wrong format (overflow of the page) at line 530, 639
  9. In the keyword section, those words should be further details explained with example
  10. In the “Keops Algorithm” section, the picture should be further details explained.
  11. In the Strategy (time-optimized) section, the figure 5 should be further increase size, because the text is so small.

Author Response

Comments to the Author

  1. Regarding the proposed hasting algorithm, lacking descriptions on contribution manner and its functionality obligated extending distinguish.
    2.      The author should provide further descriptions of the paper contributions and detailed reason why the proposed approach enhances existed works.
    3.      The general definitions of index terms should be lessened to strengthen paper to meet technical sound.
    4.      Acronym, the authors haven’t written the full meaning of word (ECUs) at line 126
    5.      Acronym, the authors haven’t written the full meaning of word (LZ) at line 165
    6.      Acronym, the authors haven’t written the full meaning of word (RLZ) at line 172
    7.      The authors haven’t forgotten remove empty of two pages at the last of Journal Paper
    8.      In the Reference section, the link of title is wrong format (overflow of the page) at line 530, 639
    9.      In the keyword section, those words should be further details explained with example
    10.     In the “Keops Algorithm” section, the picture should be further details explained.
    11.     In the Strategy (time-optimized) section, the figure 5 should be further increase size, because the text is so small.

Response: Thank you for your time and comments.

Reviewer Comment:  

Regarding the proposed hasting algorithm, lacking descriptions on contribution manner and its functionality obligated extending distinguish.

Response: We are not sure about what the reviewer is referring to. We kindly ask the reviewer to rephrase.

Reviewer Comment:  

The author should provide further descriptions of the paper contributions and detailed reason why the proposed approach enhances existed works.

Response: We have updated the Abstract, Section 2 (two paragraphs were added in the end), and the Conclusions to address this issue.

Reviewer Comment:  

The general definitions of index terms should be lessened to strengthen paper to meet technical sound.

Response: We are not sure about what index terms the reviewer is referring to. We kindly ask the reviewer to rephrase. Maybe the fact that we have added definitions for DELTA rate, compressed DELTA rate and ZIP rate covers the reviewer comment.

Reviewer Comment:  

Acronym, the authors haven’t written the full meaning of word (ECUs) at line 126

Acronym, the authors haven’t written the full meaning of word (LZ) at line 165

Acronym, the authors haven’t written the full meaning of word (RLZ) at line 172

Response: ECU (footnote pg. 4), LZ (footnote pg. 1) and RLZ (footnote pg. 1) acronyms are now explained in the text where they are first mentioned.

Reviewer Comment:  

The authors haven’t forgotten remove empty of two pages at the last of Journal Paper

Response: The pages have been removed

Reviewer Comment:  

In the Reference section, the link of title is wrong format (overflow of the page) at line 530

Response: The link has been updated to fit within page size.

Reviewer Comment:  

In the keyword section, those words should be further details explained with example

Response: We made extensive use of any of these keywords within the article.

Reviewer Comment:  

In the “Keops Algorithm” section, the picture should be further details explained.

Response: We kind ask the reviewer to specify what figures should be explained in more details.  

Reviewer Comment:  

In the Strategy (time-optimized) section, the figure 5 should be further increase size, because the text is so small.

Response: Fig. 5 has been updated such to make visible the text.

To see the updated version of the paper, please download the attached file.

Reviewer 3 Report

In this paper, a novel delta differencing algorithm that combines software updating methods is proposed. A general idea is interesting, and the presented numerical results indicate that the proposed algorithm can be interesting for the scientific community.

The introduction provides sufficient background, presents the possible applications in the automotive industry, and gives a short overview of similar algorithms. The overview of the related work is given in the next section, and the cited references are adequate, although some references could be omitted.

The original contribution is mostly related to the Section 3, where the Keops algorithm is proposed. The algorithm is based on LZ77 algorithm, and three new strategies for buffer paring are proposed.  The presented numerical results indicate that 1-to-1 strategy provides the lowest time consumption, while the brute force strategy provides the best compression.

It looks that the numerical results for Silecia 1 /2 are not presented in Figure 10. If only the packages related to the automotive industry are intended to be shown, please remove the results for mingw package, otherwise add the results for the Silencia package. The brute force strategy is not visible for Alog package, and the corresponding comment should be given in the text. 

Also, a few technical issues should be corrected before publication:

 - Figures 6 and 7 should be presented on the same page;

 - description of the lines for thee strategies in figure 10 should be placed as a legend in every particular diagram, it should not be placed on the bottom of the figures;  

 - line alignment should be corrected in line 487;

 - some statements should be written in a more precise way, e.g. in line 372, are the delta rates very high or very low?

- all references should be written according to the journal template!

Author Response

Comments to the authors

In this paper, a novel delta differencing algorithm that combines software updating methods is proposed. A general idea is interesting, and the presented numerical results indicate that the proposed algorithm can be interesting for the scientific community.

The introduction provides sufficient background, presents the possible applications in the automotive industry, and gives a short overview of similar algorithms. The overview of the related work is given in the next section, and the cited references are adequate, although some references could be omitted.

The original contribution is mostly related to the Section 3, where the Keops algorithm is proposed. The algorithm is based on LZ77 algorithm, and three new strategies for buffer paring are proposed.  The presented numerical results indicate that 1-to-1 strategy provides the lowest time consumption, while the brute force strategy provides the best compression.

Response: Thank you for your time in reviewing and writing comments about our article.

Reviewer Comment:

It looks that the numerical results for Silecia 1 /2 are not presented in Figure 10. If only the packages related to the automotive industry are intended to be shown, please remove the results for mingw package, otherwise add the results for the Silencia package.

Response: Our intention was to show the complete results on all test cases. Such cases of Silezia 1 & 2 are as interesting as the others; the omission was unintentional, and it has been corrected.

Please revisit Fig 10.

Reviewer Comment:

The brute force strategy is not visible for Alog package, and the corresponding comment should be given in the text.

Response: Brute Force and Flexy points plot almost identical lines. Same as for Replay, where 1-to-1 strategy gives same results as Brute Force and Flexy. We have explained this in the text by adding the following comment at line 384.

“Concerning the strategy, the plots in Figure 10 show that Brute Force is generally the best one if the goal is a good compression. However, there are exceptions like for Replay package, where all strategies give the same DELTA rate (the plots are superposed) or Alog package with Brute Force and Flexi behaving identically.”

Reviewer Comment:

Also, a few technical issues should be corrected before publication:

 - Figures 6 and 7 should be presented on the same page;

Response: We presented Figure 5 and Figure 6 on the same since Brute Force strategy in Figure 6 is the solution for code desynchronizations given as example in Figure 5.  

Reviewer Comment:  

- description of the lines for thee strategies in figure 10 should be placed as a legend in every particular diagram, it should not be placed on the bottom of the figures; 

Response: The legends were included in each diagram of Figure 10.

Reviewer Comment:  

- line alignment should be corrected in line 487;

Response: We are not sure about this comment. Please rephrase.

Reviewer Comment:  

 - some statements should be written in a more precise way, e.g. in line 372, are the delta rates very high or very low?

Response: The text has been updated for clarity. Al line 392, it now states:

“As the results in Table 2 show, the DELTA rate of Alog reaches even 0.15\%. This means the Delta file is less than 1\% in size when compared to the target size. It is a very good result, in line with our predictions, especially when compared to the ZIP rates of 7.96%. This shows that Keops algorithm performs extremely well when applied on high redundancy data, due to content similarities.

For mingw and swfk on the other hand, having less redundant versions, the minimum DELTA rates are 41.95% and 14.98% but still lower than the actual ZIP (Deflate) compression rates shown in Table 2.

Generally, the difference between DELTA rate and ZIP rate grows when large history buffers are used by Keops. For example, Replay shows an improvement in DELTA Rate from 22.15 to 1.33\% as compared to 75.78% ZIP Rate”

Reviewer Comment:  

- all references should be written according to the journal template!

Response: The references were written according the journal template.

To see the updated version of the paper, please download the at

Round 2

Reviewer 1 Report

In this revision, all my previous concerns have been well responded to. No further comment has been raised.

Reviewer 2 Report

English should be revised and corrected. It is highly inadequate for publishing.

Most of the errors have to do with structural, grammatical or syntax issues.

Reviewer 3 Report

Most of my comments are taken into account. The comment releated to justification of the text in one line is not so important in this moment, and it can be fixed in the proof. I can recomment the publication of the paper in the present form.